# Truly Scale-Equivariant Deep Nets with Fourier Layers

**Md Ashiqur Rahman      Raymond A. Yeh**
Department of Computer Science, Purdue University
{rahman79, rayyeh}@purdue.edu

## Abstract

In computer vision, models must be able to adapt to changes in image resolution to effectively carry out tasks such as image segmentation; This is known as scale-equivariance. Recent works have made progress in developing scale-equivariant convolutional neural networks, e.g., through weight-sharing and kernel resizing. However, these networks are not truly scale-equivariant in practice. Specifically, they do not consider anti-aliasing as they formulate the down-scaling operation in the continuous domain. To address this shortcoming, we directly formulate down-scaling in the discrete domain with consideration of anti-aliasing. We then propose a novel architecture based on Fourier layers to achieve truly scale-equivariant deep nets, i.e., absolute zero equivariance-error. Following prior works, we test this model on MNIST-scale and STL-10 datasets. Our proposed model achieves competitive classification performance while maintaining zero equivariance-error.

## 1   Introduction

Consider the task of image classification; if an object in the image is scaled (resized), then its corresponding object label should remain the same, *i.e.*, scale-invariant. Similarly, for semantic segmentation, if an object is scaled, then its corresponding mask should also be scaled accordingly, *i.e.*, scale-equivariant. Similarly, one would expect the features extracted to be scale-equivariant; see Fig. 1 for illustration. These invariant and equivariant properties are important to many computer vision tasks due to the nature of images. A photo of the same scenery can be taken from different distances, and objects in the scenes may come in different sizes. Developing representations that effectively capture this multi-resolution aspect of images has been a long-standing quest [1, 9, 11, 17, 53].

Recently, there has been a line of work on developing scale-equivariant convolutions networks [8, 13, 41, 42, 46] to more effectively learn multi-resolution features. At a high level, these works achieve scale-equivariant convolution layers through weight-sharing and kernel resizing, *i.e.*, use the "same" but resized kernel across all scales [5]. The innovation of these works is how to properly resize the kernel. For example, Bekkers [2] and Sosnovik et al. [41] formulate kernel resizing as a continuous operation and then discretize the kernel when implemented in practice. However, this discretization leads to non-negligible equivariance error. On the other hand, Worrall and Welling [46] and  Sosnovik et al. [42] directly formulate kernel resizing in the discrete domain, *e.g.*, using dilation or solving for the best kernel given a fixed scale set, and achieve low equivariance-error.

Despite these successes, we point out that the aforementioned works are not truly scale-equivariant in practice. Specifically, these works are derived using a continuous domain down-scaling operation, *i.e.*, there is no need to consider anti-aliasing. However, when performing a down-scaling on discrete space, the Nyquist theorem [23, 30] tells us that an anti-alias filter is necessary to avoid high-frequency content to alias into lower frequencies. The canonical example of aliasing is the "wagon-wheel effect", where a wheel in a video appears to be rotating slower or even in reverse from its true rotation. To

37th Conference on Neural Information Processing Systems (NeurIPS 2023).

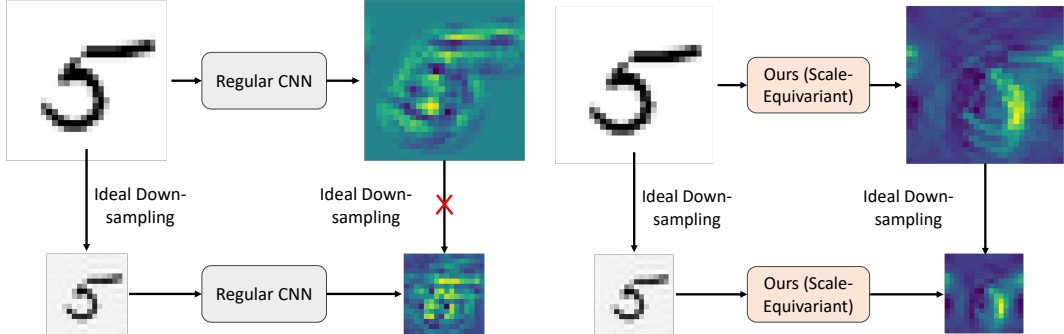

**(a)** Illustration that regular CNNs are not scale-equivariant. **(b)** Illustration that our model is scale-equivaraint.

**Figure 1.** Comparison of scale-equivariance on CNN vs. our model. For a regular CNN, the features extracted from the corresponding high/low image resolution look very different. On the other hand, downsampling the high-res feature is guaranteed to achieve the same feature obtained from the low-resolution image.

address this gap from prior work, we consider the down-scaling operation directly in the discrete domain, taking he anti-aliasing into account.

In this work, we formulate down-scaling as the ideal downsampling from signal processing [23]. We then propose a family of deep nets that are truly scale-equivariant based on this ideal downsampling. This involves rethinking all the components in the deep net, including convolution layers, non-linearities, and pooling layers.

With the developed deep net, we focus on the task of image classification. We further point out that truly scale-invariant classifiers are not desirable. A truly scale-invariant model's performance is limited by the lowest-resolution image. Instead, the more desirable property is that a high-resolution image should achieve a better performance than its corresponding low-resolution image. This motivated us to design a classifier architecture suitable for this property.

Following prior works, we conduct our experiments on the MNIST-scale [40] and STL [4] dataset. By design, our method achieves zeros scale equivariance-error both in theory and in practice. In terms of accuracy, we compare to recent scale-equivariant CNNs. We found our approach to be competitive in classification accuracy and exhibit better data efficiency in low-resource settings.

**Our contributions are as follows:**

- We formulate down-scaling in the discrete domain with considerations of anti-aliasing.

- We propose a family of deep nets that is truly scale-equivariant by designing novel scale-equivariant modules based on Fourier layers.

- We conduct extensive experiments validating the proposed approach. On MNIST and STL datasets, the proposed model achieves an absolute zero end-to-end scale-equivariance error while maintaining competitive classification accuracy.

## 2 Related Work

**Scale-equivariance and invariance.** The notation of scale-equivariance is deeply rooted in image processing and computer vision. For example, classic hand-designed scale-invariant features such as SIFT [21, 22] have made tremendous contributions to the field of computer vision. Earlier works propose to use an image or spatial pyramid to capture the multi-resolution aspect of an image [1, 9, 17] by extracting features at several scales in an efficient manner.

More recently, there have been interests in developing scale-equivariant CNN [2, 8, 13, 41, 42, 46, 54]. Based on Group-Conv [5], these works achieve scale-equivariance convolution layers through weight-sharing and kernel resizing. Different from these works, we consider the down-scaling in the discrete domain formulated as ideal downsampling from signal processing. We then develop modules that are truly scale-equivariant to enable a deep net that achieves zero equivariance-error measured from end to end. Finally, we note that there is a rich literature of equivariant deep nets [3, 5, 33, 36–38, 44, 45, 50]

with numerous applications applied to various domains, *e.g.*, sets [10, 26, 32, 34, 48, 51], graphs [6, 7, 15, 19, 20, 25, 28, 39, 49], etc. Moreover, several recent studies have also identified and tackled the issue of aliasing generated from the pooling layer to attain finer translation equivariance [35, 47, 52] and image generation [14].

**Fourier transform in neural networks.** Fourier transforms have been previously used in deep learning. For example, Mathieu et al. [27] proposes to use Fast Fourier Transform (FFT) to speed up CNN training. Fourier transform has also been used to develop network architectures, including various convolutional neural networks operating in the Fourier space [16, 31]. Recently, Fourier layers are capable of handling inputs of varying resolution, which have been employed in neural operators, facilitating applications in partial differential equations and in state space models [18, 29]. Fourier convolutions have also found success in low-level image processing tasks, *e.g.*, inpainting [43], deblurring [24]. Different from these works, we focus on developing truly scale-equivariant deep nets and leverage Fourier layers to achieve this goal.

## 3 Preliminaries

We briefly introduce and review the definition of Fourier transform, ideal downsampling, and scale-equivariance. For readability, we use 1D data to define these concepts. These ideas are extended to 2D data with multiple channels when implemented in practice.

**Discrete Fourier Transform (DFT).** Given an input vector $\mathbf{x} \in \mathbb{R}^N$, we consider $\mathcal{F} : \mathbb{R}^N \to \mathbb{C}^N$ be the discrete Fourier Transform (DFT) which has the form

$$\mathbf{X} = \mathcal{F}(\mathbf{x}) \text{ such that } \mathbf{X}[k] \triangleq \frac{1}{N} \sum_{n=0}^{N-1} \mathbf{x}[n] e^{-j\frac{2\pi}{N}kn}, \tag{1}$$

where $j$ denotes the unit imaginary number, *i.e.*, $j^2 = -1$. The index $k$ in Eq. (1) is commonly within the domain of $[0, N]$. Note that as Eq. (1) is $N$-periodic, for readability, we will use $k$ from $[-\frac{N-1}{2}, \frac{N-1}{2}]$ where $k = 0$ corresponds the lowest frequency.

The corresponding inverse DFT (IDFT) $\mathcal{F}^{-1} : \mathbb{C}^N \to \mathbb{R}^N$ is defined as

$$\mathbf{x} = \mathcal{F}^{-1}(\mathbf{X}) \text{ such that } \mathbf{x}[n] = \sum_{k=0}^{N-1} \mathbf{X}[k] e^{j\frac{2\pi}{N}kn}. \tag{2}$$

By the convolution property of DFT, the circular convolution between $\mathbf{x}$ and a kernel $\mathbf{k} \in \mathbb{R}^N$ can be represented as the element-wise multiplication in the Fourier domain, *i.e.*,

$$\mathcal{F}(\mathbf{x} \circledast \mathbf{k}) = \mathcal{F}(\mathbf{x}) \odot \mathcal{F}(\mathbf{k}) = \mathbf{X} \odot \mathbf{K}, \tag{3}$$

where $\circledast$ denotes the circular convolution and $\odot$ denotes element-wise multiplication. Unless explicitly mentioned, we will represent the input vector with lowercase letters (e.g., $\mathbf{x}$) and its corresponding DFT with uppercase letters (e.g., $\mathbf{X}$).

**Down-scaling operation.** To reduce the scale (or resolution) of a signal $\mathbf{x} \in \mathbb{R}^N$, one could perform a subsampling $\mathtt{Sub}_R$ by a factor of $R$

$$\mathtt{Sub}_R(\mathbf{x})[n] = \mathbf{x}[Rn]. \tag{4}$$

However, naively subsampling leads to aliasing. Hence, anti-aliasing is performed in a multi-rate system. In signal processing, the analysis commonly uses the ideal anti-aliasing filter $\mathbf{h}$, which zeros out all the high-frequency content, *i.e.*, its DFT $\mathbf{H} \triangleq \mathcal{F}(\mathbf{h})$ is defined as:

$$\mathbf{H}[k] = 1 \text{ if } |k| \leq \frac{N}{2R} \text{ and } 0 \text{ otherwise.} \tag{5}$$

See Fig. 2a for an illustration of the ideal anti-aliasing filter.

In this work, we define the overall down-scaling operation to be the ideal downsampling $\mathcal{D}_R$ by a factor of $R$, which performs anti-aliasing followed by a subsampling operation:

$$\mathcal{D}_R(\mathbf{x}) \triangleq \mathtt{Sub}_R(\mathbf{h} \circledast \mathbf{x}) \ \forall R < N, \tag{6}$$

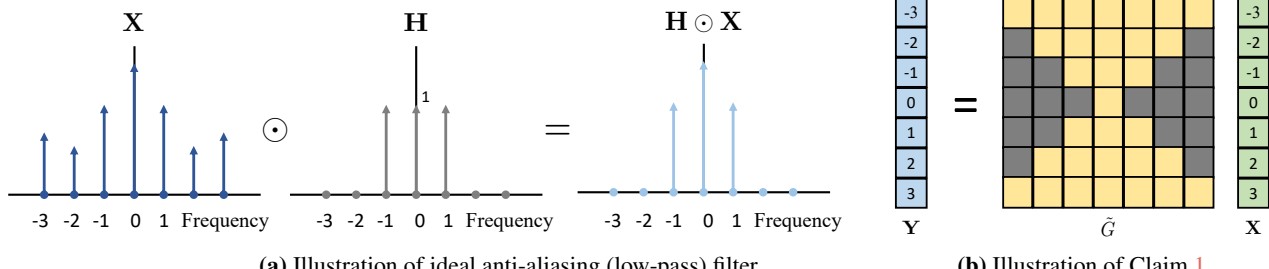

**(a)** Illustration of ideal anti-aliasing (low-pass) filter.

**(b)** Illustration of Claim 1.

**Figure 2.** In (a), we illustrate an ideal low-pass filter showing that it zeros out the high frequencies. In (b), we illustrate the structure described in Claim 1 for a linear $\tilde{G}$. The **gray regions** correspond to the value being zero.

where their DFT are related by

$$\mathcal{F}(\mathcal{D}_R(\mathbf{x})) = \mathcal{F}(\mathbf{x})\left[-N/2R : N/2R\right]. \tag{7}$$

**Scale-equivariance.** With the down-scaling operation defined, a deep net $\mathbf{g} : \{\mathbb{R}^1, \mathbb{R}^2, \ldots, \mathbb{R}^N\} \mapsto \{\mathbb{R}^1, \mathbb{R}^2, \ldots, \mathbb{R}^N\}$ is scale-equivariant if:

$$\mathbf{g}(\mathcal{D}_R(\mathbf{x})) = \mathcal{D}_R(\mathbf{g}(\mathbf{x})) \quad \forall \mathbf{x} \in \{\mathbb{R}^1, \mathbb{R}^2, \ldots, \mathbb{R}^N\} \text{ and } R < \dim(\mathbf{x}), \tag{8}$$

where $\{\mathbb{R}^1, \mathbb{R}^2, \ldots, \mathbb{R}^N\}$ represents the space of input/output signals at different scales. In this paper, we are interested in designing a family of deep nets that satisfies the equality in Eq. (8). Scale-invariance can be defined in a similar manner as

$$\mathbf{g}(\mathcal{D}_R(\mathbf{x})) = \mathbf{g}(\mathbf{x}) \quad \forall \mathbf{x} \in \{\mathbb{R}^1, \mathbb{R}^2, \ldots, \mathbb{R}^N\} \text{ and } R < \dim(\mathbf{x}). \tag{9}$$

**Fourier layer.** Given a multi-channel input vector, $\mathbf{x} \in \mathbb{R}^{C_{\text{in}} \times N}$ and kernel $\mathbf{k} \in \mathbb{R}^{C_{\text{out}} \times C_{\text{in}} \times N}$, where $C_{\text{in/out}}$ is the number of input/output channels, the circular convolution layer is defined as

$$\mathcal{F}((\mathbf{x} \circledast \mathbf{k}))[c'] = \sum_{c=1}^{C_{\text{in}}} \mathbf{X}[c] \odot \mathbf{K}[c', c], \tag{10}$$

where $\mathbf{X}$ and $\mathbf{K}$ denotes the DFT of $\mathbf{x}$ and $\mathbf{k}$ applied independently for each channel.

## 4 Approach

Our goal is to design truly scale-equivariant deep nets. To accomplish this goal, we propose scale-equivariant versions of CNN modules, including, the convolution layer, non-linearities, and pooling layers. In Sec. 4.1, we detail the operation for each of the proposed modules. In Sec. 4.2, we demonstrate how to build a classifier that is suitable for image classification with scale-equivariant features. We now explain our overarching design principle for the scale-equivariant modules.

From a frequency perspective, as reviewed in Eq. (6), the ideal downsampling operation results in the loss of higher frequency terms of the signal. In other words, if a feature's frequency terms depend on any higher frequency terms of the input, then it is not scale-equivariant, as the information will be lost after downsampling. We now formally state this observation in Claim 1.

**Claim 1.** *Let $g$ denote a deep net such that $\mathbf{y} = g(\mathbf{x})$. If this deep net $g$ can be equivalently represented as a set of functions $\tilde{G}_k : \mathbb{C}^{2k+1} \to \mathbb{C}$ such that*

$$\mathbf{Y}[k] = \tilde{G}_k(\mathbf{X}[-k:k]) \ \ \forall k \tag{11}$$

*then $g$ is scale-equivariant as defined in Eq. (8). In other words, an output's frequency terms can only have dependencies on the terms in $\mathbf{X}$ that are **even lower** in frequencies. We illustrate this structure with a linear function in Fig. 2b.*

*Proof.* We denote the deep net's input and output as $\mathbf{x}$ and $\mathbf{y}$ with corresponding DFT $\mathbf{X}$ and $\mathbf{Y}$. We denote the deep net's down-scaled input and output as $\mathbf{x}' = \mathcal{D}_R(\mathbf{x})$ and $\mathbf{y}' = g(\mathbf{x}')$ with corresponding DFT $\mathbf{X}'$ and $\mathbf{Y}'$. Now assume that $g : \mathbb{R}^n \to \mathbb{R}^n \ \ \forall n \in \{1, 2, \dots N\}$ is a deep net that satisfies Claim 1 then

$$\mathbf{Y}[k] = \tilde{G}_k(\mathbf{X}[-k : k]) \ \ \forall k \leq \frac{N}{R} \tag{12}$$

$$= \tilde{G}_k(\mathbf{X}'[-k : k]) = \mathbf{Y}'[k] \ \text{ Following the property of } \mathcal{D}_R \text{ in Eq. (7)} \tag{13}$$

Therefore, $\forall k \leq \frac{N}{R} \ \ \mathbf{Y}[k] = \mathbf{Y}'[k]$. By the definition of ideal downsampling $\mathbf{Y}' = \mathcal{D}_R(\mathbf{Y})$ concluding that $g(\mathcal{D}_R(\mathbf{x})) = \mathcal{D}_R(g(\mathbf{x}))$, *i.e.*, $g$ is scale-equivariant.

For ease of understanding, here we assume that the deep net's input and output are of the same size. A version with a more relaxed assumption is provided in Appendix Sec. A1. $\qquad\square$

## 4.1 Scale Equivariant Fourier Networks.

We now describe the proposed modules and show that they are truly scale-equivariant.

**Spatially local Fourier layer.** For computer vision, learning local features is crucial. The Fourier layer in Eq. (10) is global in nature. To efficiently learn local features, we propose a localized Fourier layer where we constrain the degree of freedom in the kernel $\mathbf{K}$ such that the respective spatial kernel $\mathbf{k}$ is spatially localized.

Let $\mathbf{k} \in \mathbb{R}^d$ and $\mathbf{k}^l \in \mathbb{R}^l$ to be $d$ and $l$ dimensional kernel such that $\mathbf{k}[i] = \mathbf{k}^l[i]$ if $i < l$ otherwise $0$, *i.e.*, $\mathbf{k}$ is spatially local to have a receptive field of size $l$. We denote $\mathbf{K}$ and $\mathbf{K}^l$ be the DFT of the kernel $\mathbf{k}$ and $\mathbf{k}^l$ respectively. We claim that $\mathbf{K}$ can be written as

$$\mathbf{K}[p] = \frac{1}{d} \sum_{m=-\frac{l}{2}}^{\frac{l}{2}} \left( \mathbf{K}^l[m] \sum_{n=0}^{l-1} e^{-2\pi j n (\frac{p}{d} - \frac{m}{l})} \right). \tag{14}$$

From Eq. (14), instead of modeling all the degrees of freedom in $\mathbf{K}$, we will directly parameterize $\mathbf{K}^l$ to enforce the learned kernel to be localized spatially. We defer the proof to the Appendix Sec. A2.

> **Claim 2.** *The spatially local Fourier layer is scale-equivariant.*

*Proof.* The kernel $\mathbf{k}$ has a corresponding DFT $\mathbf{K}$. As reviewed, a circular convolution between $\mathbf{k}$ and input $\mathbf{x}$ can be expressed as

$$\mathbf{Y}[k] = \mathbf{K}[k] \odot \mathbf{X}[k] \ \forall k. \tag{15}$$

Observe that $\mathbf{X}[k]$ is a subset of $\mathbf{X}[-k : k]$, *i.e.*, Claim 1 is satisfied. $\qquad\square$

**Scale-equivariant non-linearity ($\sigma_\mathbf{s}$).** Element-wise non-linearities, *e.g.*, ReLU, in the spatial domain are generally not scale-equivariant under the ideal downsampling operation $\mathcal{D}_R$. While applying element-wise non-linearity in the frequency domain is scale-equivariant, this strategy empirically leads to degraded performance on classification tasks. To address this, we propose a scale-equivariant non-linearity $\sigma_\mathbf{s}$ in the spatial domain.

Given a non-linearity $\sigma$, *e.g.*, ReLU, we construct a corresponding scale-equivariant version $\sigma_\mathbf{s}$ that satisfies Claim 1. Let $\mathbf{x} \in \mathbb{R}^N$ and $\mathbf{y} \in \mathbb{R}^N$ to denote the input and output of $\sigma_\mathbf{s}$. We define scale-equivariant non-linearity $\sigma_\mathbf{s}(\mathbf{x}) = \mathcal{F}^{-1}(\mathbf{Y})$ where $\mathbf{Y}$ takes the following form:

$$\mathbf{Y}[k] = \begin{cases} \mathcal{F}\Big(\sigma \circ \mathcal{F}^{-1}(\mathbf{X}[0])\Big)[0], & k = 0 \\ \mathcal{F}\Big(\sigma \circ \mathcal{F}^{-1}(\mathbf{X}[-1 : 1])\Big)[1], & k = 1 \\ \vdots & \vdots \\ \mathcal{F}\Big(\sigma \circ \mathcal{F}^{-1}(\mathbf{X}[-|k| : |k|])\Big)[k], & k = k \\ \vdots & \vdots \end{cases} \tag{16}$$

$\forall |k| \leq \frac{N}{2}$ and $\mathbf{X}$ denotes the DFT of the input, *i.e.*, $\mathcal{F}(\mathbf{x})$. In practice, we choose $\sigma$ to be ReLU in our implementation.

Next, it is generally computationally expensive to achieve equivariance over all scales. In practice, we only enforce a set of scales for which we want to achieve equivariance, which can be denoted in terms of corresponding resolutions as $\mathcal{R} = (m, \dots N)$ with $\mathcal{R}[i] < \mathcal{R}[i+1]$. To achieve scale-equivariant non-linearity over the scales of $\mathcal{R}$, $\sigma_{\mathbf{s}} = \mathcal{F}^{-1}(\mathbf{Y})$ can be efficiently computed as

$$\mathbf{Y}[k] = \mathcal{F}\Big(\sigma \circ \mathcal{F}^{-1}(\mathbf{X}\big[-\frac{\mathcal{R}'[i]}{2} : \frac{\mathcal{R}'[i]}{2}\big])\Big)[k] \quad \text{for the } i \text{ s.t. } \frac{\mathcal{R}'[i-1]}{2} < |k| \leq \frac{\mathcal{R}'[i]}{2}. \quad (17)$$

Here, the ordered set $\mathcal{R}' = \mathcal{R} \cup \{0\}$. By Eq. (17), all the Fourier coefficients $k$ between any two consecutive resolutions in $\mathcal{R}$, *i.e.*, $\mathcal{R}[i-1]/2 < |k| \leq \mathcal{R}[i]/2$ can be computed by a single Fourier transform pair.

**Scale-equivariant pooling.** Pooling operation is crucial for deep nets' scalability to larger images and datasets as they make the network more memory and computationally efficient. Commonly used pooling operations are max/average pooling, which reduces the input size by the factor of its window size $w$ and is not scale-equivariant. To address this, we propose scale-equivariant pooling $\texttt{Pool}_{\mathbf{s}}^w$.

Let $\texttt{Pool}^w$ denote a max/average pooling operation with a window size $w$ and $\texttt{Pool}^w : \mathbb{R}^d \to \mathbb{R}^{\frac{d}{w}}$. We define scale-equivariant pooling operation $\texttt{Pool}_{\mathbf{s}}^w : \mathbb{R}^d \to \mathbb{R}^{\frac{d}{w}}$ mapping from $\mathbf{x}$ to $\mathbf{y}$ where $\mathbf{y} = \mathcal{F}^{-1}(\mathbf{Y})$ follows

$$\mathbf{Y}[k] = \mathcal{F}\big(\texttt{Pool}^w(\mathcal{F}^{-1}(\mathbf{X}\big[-w|k| : w|k|\big])))\big)[k] \quad \forall k \leq \frac{d}{2w}. \quad (18)$$

Observe that this pooling layer satisfies Claim 1 by construction. Similar to non-linearity, we can enforce the equivariance over the set $\mathcal{R}$ following the same formulation in Eq. (17).

Note that as pooling reduces the size of the output by a factor $w$, the operation is only scale-equivariant at every $w^{\text{th}}$ resolution. When the input size is not a multiple of $w$ there is a truncation of the input.

**Time Complexity.** We now provide the time complexity of our scale-equivariant Fourier layer and compare it with standard group convolutions. Let's consider a 1D signal of length $N$ and a kernel of length $K$. Our proposed model involves:

- A transformation of local filter to global with time complexity $O(KN)$
- A convolution using Fourier transform with time complexity $O(N log(N))$
- Our scale equivariant non-linearity depends on the size of the group. Let $A$ be the set of group actions. The time complexity of the proposed scale-equivariant non-linearity is $O(|A|N \log(N))$, where $|A|$ denotes the cardinality of the set $A$.

So, the time complexity for each layer becomes

$$O(|A|N \log(N) + KN).$$

As a comparison, the time complexity of regular group convolution is $O(KN|A|)$ in the first layer and $O(KN|A|^2)$ for all intermediate layers, assuming the cost of group action is a negligible constant [12].

Considering the time complexity of the intermediate layers of group convolutions, our proposed method is more efficient when

$$|A|N \log(N) + KN < |A|^2 KN \implies \log(N) + \frac{K}{|A|} < |A|K.$$

So, when $K << |A|$ and $log(N) < |A|K$, i.e., assuming the set of group actions of moderate size, then our method is faster than group convolutions.

Modern GPUs are specifically optimized for regular convolution operations that can be performed in place. In contrast, the FFT algorithm does not fully capitalize on GPUs' advantages, primarily due to unique memory access patterns and moderate arithmetic intensities. Consequently, our approach is unable to harness the full potential of GPUs. When executed on a GPU, regular group convolutions implemented as standard convolutions might exhibit comparable or even shorter running times than our approach.

## 4.2 Classifier for equivariant features

A truly scale-invariant, defined in Eq. (9), the model's performance is limited by the lowest resolution as the prediction needs to be the same. In the extreme, the prediction can only depend on a single mean pixel. Instead of invariance, we believe that it is more desirable to ensure that a high-resolution image achieves a better performance than its down-scaled version, *i.e.*, the performance is scale "consistent". To achieve this property, we propose a suitable classifier architecture and training scheme.

**Classifier.** In order to enforce scale-consistency, we need a classifier that outputs a prediction per scale. This motivated the following proposed architecture. Let $c$ be a classifier with $M$ classes where $\hat{\boldsymbol{y}} = c \circ g(\mathbf{x}) \in \mathbb{R}^{|\mathcal{R}(\mathbf{x})| \times M}$. $\mathcal{R}(\mathbf{x})$ is defined as the set of resolutions smaller than the input resolution in the considered scales $\mathcal{R}$. *i.e.*, $\mathcal{R}(\mathbf{x}) = \{k : k \leq \dim(\mathbf{x}) \text{ and } k \in \mathcal{R}\}$. Here, $g$ is a scale-equivariant deep-net that extracts features $\phi = g(\mathbf{x})$ with corresponding DFT of $\Phi$. Our proposed classifier has the form:

$$\hat{\boldsymbol{y}}[k] = \texttt{MLP} \circ \texttt{Pool}\left( \mathcal{F}^{-1}\left( \texttt{Pad}_{\texttt{N}}(\Phi[-\frac{|k|}{2} : \frac{|k|}{2}]) \right) \right) \ \forall k \in \mathcal{R}(x) \tag{19}$$

where $\texttt{Pad}_{\texttt{N}}$ is a Fourier padding operation that symmetrically pads zero to either side of the DFT to a fixed size N, $\texttt{Pool}$ is a spatial pooling operation and $\texttt{MLP}$ maps the pooled feature to the predicted logits $\hat{\boldsymbol{y}}[k]$ for each scale; Note the MLP is shared across all scales. As we are sharing the MLP, we need to ensure that the input sizes are identical. Hence, we padded the features $\Phi$ to a fixed size. Finally, at test-time, we use the output from $\hat{\boldsymbol{y}}[\dim(\mathbf{x})]$ to make a prediction.

**Training.** Given a dataset $\mathcal{T} = \{(\mathbf{x}, y)\}$, we train our model using the sum of two losses. The first term is a standard sum of cross entropy loss $\mathcal{L}$ over the scales:

$$\sum_{k \in \mathcal{R}(\mathbf{x})} \mathcal{L}(\hat{\boldsymbol{y}}[k], y). \tag{20}$$

The second term is a consistency loss to encourage the performance of high-resolution to be better than the low-resolution:

$$\sum_{k \in \mathcal{R}(\mathbf{x})} \max\left( \mathcal{L}(\hat{\boldsymbol{y}}[k], y) - \mathcal{L}(\hat{\boldsymbol{y}}[k-1], y), 0 \right). \tag{21}$$

This is a hinge loss that penalizes the model when the cross entropy loss $\mathcal{L}$ on high-resolution features (larger $k$) is greater than that of the low-resolution features (smaller $k$).

## 5 Experiments

To study the effectiveness of our model, we conduct experiments on two benchmark datasets, MNIST-scale [40] and STL10 [4], following our theoretical setup using ideal downsampling. In this case, the theory exactly matches practice, and our approach achieves perfect scale-equivariance. We also conduct experiments comparing the models' generalization to unseen scales and data efficiency. Finally, we conduct experiments using a non-ideal anti-aliasing filter in down-scaling. Under this setting, our model no longer achieves zero scale equivariance-error. However, we are interested in how the models behave under this mismatch in theory and practice.

**Evaluation metrics.** To evaluate task performance, we report classification accuracy. Next, we introduce a metric to measure the scale-consistency. Given a sample from the test set, we check whether the cross entropy loss is less than or equal to the classification loss of its down-scaled version. We compute this as a percentage over the dataset and report the scale-consistent rate defined as:

$$\text{Scale-Con.} = \frac{1}{|\mathcal{T}|} \sum_{(\mathbf{x},y) \in \mathcal{T}} \mathbb{E}_r\left( \mathbf{1}\left[ \mathcal{L}(\mathbf{x}, y) \leq \mathcal{L}(\mathcal{D}_r(\mathbf{x}), y) \right] \right), \tag{22}$$

where $r$ is uniformly sampled over the set of scales for which we want to achieve equivariance and $\mathbf{1}$ denotes the indicator function.

Finally, we quantify the equivariance-error over the final feature map given by a fully trained model on the dataset. The equivariance-error (Equi-Err.) is defined as

$$\text{Equi-Err.} = \frac{1}{|\mathcal{T}||\mathcal{S}|} \sum_{\mathbf{x} \in \mathcal{T}} \sum_{r \in \mathcal{R}} \frac{\|g(\mathcal{D}_r(\mathbf{x})) - \mathcal{D}_r(g(\mathbf{x}))\|_2^2}{\|g(\mathcal{D}_r(\mathbf{x}))\|_2^2}. \tag{23}$$

**Table 1.** Accuracy of different models on MNIST-scale (ideal downsampling) with all scales.

| Models | Acc.↑ | Scale-Con.↑ | Equi-Err.↓ |
|---|---|---|---|
| CNN | 0.9737 | 0.6621 | - |
| Per Res. CNN | 0.9388 | 0.0527 | - |
| SESN | 0.9791 | 0.6640 | - |
| DSS | 0.9731 | 0.6503 | - |
| SI-CovNet | 0.9797 | 0.6425 | - |
| SS-CNN | 0.9613 | 0.3105 | - |
| DISCO | 0.9856 | 0.5585 | 0.44 |
| Fourier CNN | 0.9713 | 0.2421 | 0.28 |
| Ours | **0.9889** | **0.9716** | **0.00** |

**Table 2.** Accuracy of different models on MNIST-scale (ideal downsampling) with missing scales.

| Models | Acc.↑ | Scale-Con.↑ | Equi-Err.↓ |
|---|---|---|---|
| CNN | 0.9842 | 0.7617 | - |
| Per Res. CNN | 0.9763 | 0.3594 | - |
| SESN | 0.9892 | 0.8339 | - |
| DSS | 0.9884 | 0.8105 | - |
| SI-CovNet | 0.9878 | 0.6621 | - |
| SS-CNN | 0.9870 | 0.3593 | - |
| DISCO | **0.9914** | 0.5371 | 0.35 |
| Fourier CNN | 0.9820 | 0.1250 | 0.23 |
| Ours | 0.9888 | **0.9366** | **0.00** |

Here, $\mathcal{R}$ is the set of all scales over which we enforce equivariance. We report the average equivariance error over the samples of the test set $\mathcal{T}$. We note that this equivariance-error *differs* from the one reported by Sosnovik et al. [42] where they measured the error for the "scale-convolution with weights initialized randomly." Contrarily, we measure the equivariance error from *end-to-end* over *trained* models, which more closely matches how the models are used in practice.

**Baselines.** Following prior works in scale-equivariant neural networks [41, 42] we compare to baselines: DISCO [42], SI-ConvNet [13], SS-CNN [8], DSS [46], and SESN [41]. For the baseline, we follow the architecture and training scheme provided by Sosnovik et al. [42]. We also prepared three additional baseline models: (a) standard CNN, (b) Per Res, CNN where we train a separate CNN for each resolution in the training set, and (c) Fourier CNN [18] which utilizes Fourier layers.

### 5.1 MNIST-scale (Ideal downsampling)

**Experiment setup.** We create the MNIST-scale dataset following the procedure in prior works [13, 42]. Each image in the original MNIST dataset is randomly downsampled with a factor of $\sim [\frac{1}{0.3} - 1]$, such that every resolution from $8 \times 8$ to $28 \times 28$ contains an equal number of samples. As the baseline models (except the Fourier CNN) can not handle images of different resolutions, following prior works, lower-resolution images are zero-padded to the original resolution. We do not need to pad the input for our model and Fourier CNN. We used 10k, 2k, and 50k for training, validation, and test set samples. For this experiment, we enforce equivariance over scales that correspond to the discrete resolutions of $\mathcal{R} = \{8, \ldots, 28\}$.

**Implementation details.** For the baselines and CNN, we follow the implementation, hyperparameters, and architecture provided in prior works [41, 42]. For Per Res. CNN, we train a separate CNN for each resolution. Each of these CNNs uses the architecture of baseline CNN. For Fourier CNN, we use the Fourier block introduced in the Fourier Neural operator [18]. Inspired by their design, we use $1 \times 1$ complex convolution in the Fourier domain along with the scale-equivariant convolution. We follow the baseline for all training hyper-parameters, except we included a weight decay of 0.01.

**Results.** In Tab. 1, we report the accuracy of the MNIST-scale dataset. We observe that our approach achieved zero equivariance error and the highest accuracy. While all models achieve similar accuracy, there is a more notable difference in the scale consistency rate. This means that our model properly captures the additional information that comes with increased resolution.

**Generalization to unseen scales.** We study the generalization capabilities of the scale-equivariant modes to unseen scales; we train them on a dataset with 10k full resolution $(28 \times 28)$ MNIST images and test on 50k samples of MNIST-scale, *i.e.*, containing different scales. For the baselines, we added random scaling argumentation during training. In Tab. 2, we observe that our model can guarantee zero equivariance error even for the unseen scales and achieves comparable performance to baselines trained with data augmentation.

**Data efficiency.** We also conduct experiments studying the data efficiency of the different models. Following the same setup as MNIST-scale, we train the models on limited training examples, 5k, 2.5k, and 1k, of different resolutions and test on 50k samples across all resolutions. In Tab. 3, we observe that our model is more data efficient than the baselines. DISCO achieves the second-best

**Table 3.** MNIST-scale accuracy with different numbers of training samples.

| Models / # Samples | 5000 | 2500 | 1000 |
|---|---|---|---|
| CNN | 0.9432 | 0.9389 | 0.8577 |
| Per Res. CNN | 0.9118 | 0.8392 | 0.5815 |
| DISCO | 0.9794 | 0.9665 | 0.9457 |
| SESN | 0.9638 | 0.9402 | 0.9207 |
| SI-CovNet | 0.9641 | 0.9437 | 0.9280 |
| SS-CNN | 0.9477 | 0.9259 | 0.9176 |
| DSS | 0.9654 | 0.9401 | 0.9281 |
| Fourier CNN | 0.9567 | 0.9419 | 0.8910 |
| Ours | **0.9835** | **0.9767** | **0.9606** |

**Table 4.** The classification accuracy of different models on STL10-scale dataset.

| Models | Acc.↑ | Scale-Con.↑ | Equi-Err.↓ |
|---|---|---|---|
| Wide ResNet | 0.5596 | 0.2916 | 0.16 |
| SESN | 0.5525 | 0.4166 | 0.04 |
| DSS | 0.5347 | 0.1979 | 0.02 |
| SI-CovNet | 0.5588 | 0.2187 | 0.03 |
| SS-CNN | 0.4788 | 0.1979 | 1.82 |
| DISCO | 0.4768 | 0.3541 | 0.06 |
| Fourier CNN | 0.5844 | 0.2812 | 0.19 |
| Ours | **0.7332** | **0.6770** | **0.00** |

**Table 5.** Ablation on consistency loss.

| | w/ consistency | | w/o consistency | |
|---|---|---|---|---|
| # Samples | Acc.↑ | Scale-Con.↑ | Acc.↑ | Scale-Con.↑ |
| 5000 | 0.9835 | 0.9296 | 0.9831 | 0.9150 |
| 2500 | 0.9767 | 0.8906 | 0.9755 | 0.8633 |
| 1000 | 0.9606 | 0.8183 | 0.9599 | 0.8144 |

performance. We also see that Per Res. CNN suffers the most when trained with fewer data points, as it trains a separate CNN for each scale and does not share parameters across different scales.

**Ablation.** We perform an ablation on the consistency loss in Eq. (21) over different training set sizes. From Tab. 5, we can observe that the consistency loss improves the accuracy of our model as well as the scale-consistency. This result validates the effectiveness of the proposed consistency loss.

### 5.2 STL10-scale (Ideal downsampling)

**Experiment setup.** Following the same procedure as the MNIST-scale dataset, we create the STL10-scale dataset. Each image of the dataset is randomly scaled with a randomly chosen downsampling factor between $[1-2]$ such that every resolution from 48 to 97 contains an equal number of samples. We use 7k, 1k, and 5k samples in our training, validation, and test set. For the baseline models, we again zero-pad the downsampled images to the original size.

**Implementation details.** For the baseline models, we use the Wide ResNet as the CNN baseline following prior work [41, 42]. For Fourier CNN, we use six Fourier blocks followed by a two-layered MLP. For our model, we use six scale-equivariant Fourier blocks followed by a two-layer MLP. All of the models are trained for 250 epochs with Adam optimizer with an initial learning rate of 0.01. The learning rate is reduced by a factor of 0.1 after every 100 epoch. For scalability, we consider achieving equivariance over scales that correspond to the discrete resolutions in the set $\mathcal{R} = \{48 \leq 48 + i \times 8 \leq 97\} \ \forall i \in \{0, 1, 2, \dots\}$.

**Results.** In Tab. 4, we observe that our model achieves zero equivariance error with higher accuracy and scale consistency over the baselines. As the baseline models accept a fixed-sized input, the downsampled images are zero-padded following prior work's preprocessing on MNIST-scale. Note, MNIST images have a uniform black background, and zero-padding does not create artifacts. However, for colored images with diverse backgrounds, such as STL-10, any padding scheme to resize the image will cause artifacts. We believe this artifact hurts the performance of baseline models on the STL10-scale dataset. However, it is unclear whether there is a more suitable padding strategy.

### 5.3 MNIST-scale (Non-ideal downsampling)

Ideal interpolation suffers from artifacts known as the ringing effect caused by *Gibbs phenomenon* [23]; see the down-scaled image in Fig. 3a. In practice, a non-ideal low-pass filter will be used instead. Taking this into consideration, we conduct the experiments using a more commonly used anti-aliasing scheme with a Gaussian blur instead of the ideal low-pass filter.

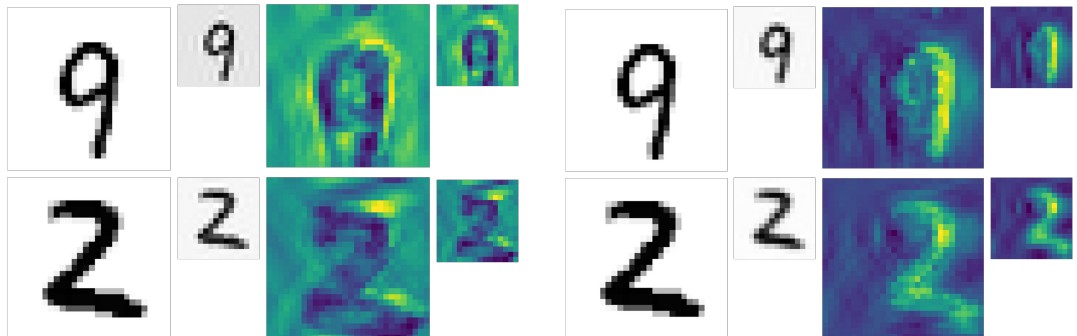

**(a)** Learned feature from ideal downsampling.      **(b)** Learned feature for non-ideal downsampling.

**Figure 3.** Feature visualization for ideal and non-ideal downsampling settings. In both settings, our model seems to learn spatially local features such as digit contour and edges.

**Table 6.** The accuracy of different models on MNIST-scale (non-ideal downsampling).

| Models | Acc.↑ | Scale Con.↑ | Equi. Err.↓ |
|---|---|---|---|
| CNN | 0.9642 | 0.1033 | - |
| Per Res. CNN | 0.9450 | 0.0742 | - |
| SESN | 0.9710 | 0.6666 | - |
| DSS | 0.9772 | 0.5716 | - |
| SI-CovNet | 0.9694 | 0.4453 | - |
| SS-CNN | 0.9670 | 0.3144 | - |
| DISCO | 0.9830 | 0.4500 | 0.63 |
| Fourier CNN | 0.9745 | 0.1716 | 0.29 |
| Ours | **0.9880** | **0.9760** | **0.05** |

**Experiment details.** We follow the same experimental setup and training scheme as in MNIST-scale with the ideal downsampling experiment. The only difference is that we use a Gaussian kernel to perform anti-aliasing.

**Results.** From Tab. 6, we observe that our model achieves higher classification accuracy and Scale consistency. Importantly, our model achieves lower equivariance error than the baseline despite the gap in the theory of non-ideal downsampling.

## 6 Conclusion

We propose a family of scale-equivariant deep nets that achieve zero equivariance error measured from end to end. We formulate down-scaling in the discrete domain with proper consideration of anti-aliasing. To achieve scale-equivariance, we design novel modules based on Fourier layers, enforcing that the lower frequency content of output does not depend on the higher frequency content of the input. Furthermore, we motivated the scale-consistency property that the performance of higher-resolution input should be better than that of the lower resolution and designed a suitable classifier architecture. Empirically, our approach achieves competitive accuracy on image classification tasks, with improved scale consistency and lower equivariance-error compared to baselines. Similar to other equivariant methodologies, defining consistent scales or group actions to achieve equivalence before constructing the model is crucial. Moreover, a common challenge all equivariant and invariant techniques face is the significant demands on memory and computational resources. In our upcoming research, we plan to enhance our approach by applying it to high-resolution image datasets and dense prediction tasks, such as instance segmentation.

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
