# Appendix:

The appendix is organized as follows:

- In Sec. A1, we provide a generalization of Claim 1 and its the complete proof.
- In Sec. A2, we provide the complete proof for Eq. (14).
- In Sec. A3, we provide additional ablations and experimental results.
- In Sec. A4, we provide additional implementation details.

## A1   Generalization of Claim 1

In the main paper, we provide proof where we assume that the input and output of the deep net are of the same size. Here, we provide a generalization without such an assumption.

> **Claim 3.** *Let $g$ denote a deep net such that $\mathbf{y} = g(\mathbf{x})$ where $\mathbf{x} \in \mathbb{R}^m$, $\mathbf{y} \in \mathbb{R}^{\frac{m}{a}}$, and $\frac{m}{a}$ is an integer $\forall m \in \mathbb{N}$ and $a \geq 1$. If this deep net $\mathbf{g}$ can be equivalently represented as a set of functions $\tilde{G}_k : \mathbb{C}^{2ak+1} \to \mathbb{C}$ such that*
> $$\mathbf{Y}[k] = \tilde{G}_k(\mathbf{X}[-ak : ak]) \ \ \forall k \tag{A24}$$
> *then $g$ is scale-equivariant as defined in Eq. (8) for all scales that creates resolutions at a multiple of $a$ after scaling the input vector $\mathbf{x}$.*

*Proof.* We denote the deep net's input and output as $\mathbf{x}$ and $\mathbf{y}$ with corresponding DFT $\mathbf{X}$ and $\mathbf{Y}$. We further denote that the deep net's down-scaled input and output as $\mathbf{x}' = \mathcal{D}_R(\mathbf{x})$ and $\mathbf{y}' = g(\mathbf{x}')$ with corresponding DFT $\mathbf{X}'$ and $\mathbf{Y}'$.

Given a deep net $g : \mathbb{R}^n \to \mathbb{R}^{\frac{n}{a}} \ \ \forall n \in \{1, 2, \dots N\}$ is a deep net that satisfies Claim 3 then

$$\mathbf{Y}[k] \quad = \tilde{G}_k(\mathbf{X}[-ak : ak]) \ \ \forall k \leq \frac{N}{R} \tag{A25}$$

$$= \tilde{G}_k(\mathbf{X}'[-ak : ak]) = \mathbf{Y}'[k] \ \ \text{Following the property of } \mathcal{D}_R \text{ in Eq. (7)} \tag{A26}$$

Therefore, $\forall k \leq \frac{N}{R} \ \ \mathbf{Y}[k] = \mathbf{Y}'[k]$. By the definition of ideal downsampling $\mathbf{Y}' = \mathcal{D}_R(\mathbf{Y})$, $g(\mathcal{D}_R(\mathbf{x})) = \mathcal{D}_R(g(\mathbf{x}))$ concluding that $g$ is scale-equivariant. Note that $\frac{n}{a}$ needs to be an integer for $\mathbb{R}^{\frac{n}{a}}$ to be a valid vector, *i.e.*, the resolution of $\mathbf{x}$ needs to be a multiple of $a$. $\qquad \square$

## A2   Proof for Spatially Localized Fourier Layer in Eq. (14)

In the main paper, we introduced the parameterization for a spatially localized Fourier layer:

$$\mathbf{K}[p] = \frac{1}{d} \sum_{m=-\frac{l}{2}}^{\frac{l}{2}} \left( \mathbf{K}^l[m] \sum_{n=0}^{l-1} e^{-2\pi j n(\frac{p}{d} - \frac{m}{l})} \right). \tag{14}$$

We now provide the derivation.

*Proof.* From definition of DFT, $\mathbf{k}$ can be written as

$$\mathbf{K}[p] \quad = \frac{1}{d} \sum_{n=0}^{d-1} \mathbf{k}[n] e^{-2j\pi \frac{np}{d}} = \frac{1}{d} \sum_{n=0}^{d-1} \mathbf{k}^l[n] e^{-2j\pi \frac{np}{d}} \ \ \text{(Ignorig the 0 elements)} \tag{A27}$$

$$= \frac{1}{d} \sum_{n=0}^{d-1} \sum_{m=-\frac{l}{2}}^{\frac{l}{2}} \mathbf{K}^l[m] e^{2j \frac{mn}{l}} e^{-2j\pi \frac{np}{d}} \ \ \text{(Using DFT of kernel } \mathbf{k}^l) \tag{A28}$$

$$= \frac{1}{d} \sum_{m=-\frac{l}{2}}^{\frac{l}{2}} \left( \mathbf{K}^l[m] \sum_{n=0}^{l-1} e^{-2\pi j n(\frac{p}{d} - \frac{m}{l})} \right) \ \ \text{(Exchanging the Sum)} \tag{A29}$$

Finally, the Geometric series $\sum_{n=0}^{l-1} e^{-2\pi jnq}$ can be expressed as

$$\sum_{n=0}^{l-1} e^{-2\pi jnq} = e^{-jq\frac{l-1}{2}} \frac{\sin(l\frac{q}{2})}{\sin(\frac{q}{2})} \text{ where } \lim_{q\to 0} \frac{\sin(l\frac{q}{2})}{\sin(\frac{q}{2})} = l. \tag{A30}$$

$\square$

## A3   Additional Results

**Additional ablations.** We conduct additional ablation studies for the proposed spatially local Fourier Layer and scale-equivariant ReLU and report the results in Tab. A1. We observe that there is a drop in accuracy of 0.5% when not using spatially local Fourier layers and that the equivariance error greatly increases if we do not use our proposed scale-equivariant ReLU.

**Table A1.** Ablation of Spatially localized Fourier filter and Scale-equivariant non-linearity

| Models | Acc.↑ | Scale-Con.↑ | Equi-Err.↓ |
|---|---|---|---|
| Ours | 0.9889 | 0.9716 | 0.00 |
| Ours w/o Local Filter | 0.9835 | 0.9628 | 0.00 |
| Ours w/o Scale-equi. ReLU | 0.9897 | 0.9492 | 7.32 |
| Fourier CNN | 0.9713 | 0.2421 | 0.28 |

**Ablation on baseline's preprocessing (Zero-padding vs. ideal upsampling).** In the main paper, all the baselines use zero-padding to pre-process images at different resolutions to the same size following prior works. However, we suspect that zero-padding on color images may hurt model performance. In Tab. A2, we provide additional experimental results by performing an ideal upsampling for the baselines. We observe that there are improvements in accuracy for the baseline models. However, our proposed model still achieves the best accuracy with the lowest equivariance-error.

**Table A2.** The classification accuracy of different models on STL10-scale dataset with **ideal** downsampling. For baseline models, images at different scales are resized via an **ideal upsampling operation**.

| Models | Acc.↑ | Scale-Con.↑ | Equi-Err.↓ |
|---|---|---|---|
| Wide ResNet | 0.6040 | 0.4791 | 0.20 |
| SESN | 0.6428 | 0.5629 | 0.08 |
| DSS | 0.6131 | 0.6562 | 0.02 |
| SI-CovNet | 0.6722 | 0.3854 | 0.03 |
| SS-CNN | 0.3246 | 0.5833 | 0.04 |
| DISCO | 0.5670 | 0.4791 | 0.05 |
| Fourier CNN | 0.5844 | 0.2812 | 0.19 |
| Ours | **0.7332** | **0.6770** | **0.00** |

In Tab. A3 and Tab. A4, we report the same ablation of zero-padding vs. ideal upsampling for the baselines. For this non-ideal downsampling setting, we observe that DSS has the lowest equivariance error, with ours achieving the second best. Our model achieves the highest accuracy out of all the models.

**Ablations of the effect of different types of padding on baselines**
In Table A5, we present the classification accuracy of various baselines on the STL1-scale dataset, employing ideal-downsampling and padding techniques, including Replicate, Circular, and Reflect.

## A4   Additional implementation details

Please refer to *our attached code* in the supplementary materials for more implementation details. Below, we briefly describe the model architectures. For the MNIST-scale dataset, the spatially localized Fourier layers use a locality size of $7 \times 7$ and $11 \times 11$. For the SLT10-scale dataset, we use a special locality of size $5 \times 5$ for all the localized Fourier layers. In both MNIST-scale and STL10-scale

**Table A3.** The classification accuracy of different models on STL10-scale dataset with **non-ideal** downsampling. For baseline models, images at different scales are resized via **zero-padding**.

| Models | Acc.↑ | Scale-Con.↑ | Equi-Err.↓ |
|---|---|---|---|
| Wide ResNet | 0.4456 | 0.3229 | 1.08 |
| SESN | 0.5155 | 0.4687 | 0.07 |
| DSS | 0.4756 | 0.3645 | **0.03** |
| SI-CovNet | 0.5234 | 0.3958 | 0.07 |
| SS-CNN | 0.3418 | 0.2187 | 1.72 |
| DISCO | 0.5125 | 0.4479 | 0.12 |
| Fourier CNN | 0.5357 | 0.5312 | 0.20 |
| Ours | **0.7262** | **0.5624** | 0.06 |

**Table A4.** The classification accuracy of different models on STL10-scale dataset with **non-ideal** downsampling. For baseline models, images at different scales are resized via an **ideal upsampling operation.**

| Models | Acc.↑ | Scale-Con.↑ | Equi-Err.↓ |
|---|---|---|---|
| Wide ResNet | 0.5952 | 0.4791 | 0.75 |
| SESN | 0.6312 | 0.5208 | 0.08 |
| DSS | 0.6126 | 0.5208 | **0.03** |
| SI-CovNet | 0.6337 | 0.4062 | 0.04 |
| SS-CNN | 0.4855 | 0.3854 | 0.05 |
| DISCO | 0.5191 | 0.4687 | 0.04 |
| Fourier CNN | 0.5357 | 0.5312 | 0.20 |
| Ours | **0.7262** | **0.5624** | 0.06 |

**Table A5.** The classification accuracy of different Scale-equivariant baseline models on STL10-scale dataset with different padding strategies.

| Models | Replicate | Circular | Reflect |
|---|---|---|---|
| SESN | 0.64 | 0.65 | 0.50 |
| DSS | 0.61 | 0.48 | 0.49 |
| SI-CovNet | 0.54 | 0.63 | 0.63 |
| SS-CNN | 0.47 | 0.50 | 0.50 |
| DISCO | 0.60 | 0.52 | 0.44 |

experiments, we use a 2D max-pooling layer before passing the flattened scale-equivariant spatial feature to the MLP. For the scale-equivariant non-linearity, we also apply instance normalization before applying point-wise non-linearity. All the models are trained on a single NVIDIA RTX 3090.