# OpenReview forum: "Truly Scale-Equivariant Deep Nets with Fourier Layers"
_NeurIPS.cc/2023/Conference — NeurIPS 2023 poster_

### Official Review · Reviewer_JmYY · 2023-07-05

**Soundness:** 4 excellent
**Presentation:** 3 good
**Contribution:** 3 good
**Rating:** 7
**Confidence:** 4

**Summary:**

The authors of this study introduce architectural modifications to the conventional CNN framework by incorporating Fourier layers. Their aim is to achieve scale-equivariance, addressing the anti-aliasing problem that occurs when images are downscaled in the continuous domain. To tackle this issue, the authors propose a discrete domain downsampling formulation and propose corresponding changes to the network architecture. Through experiments on a few datasets, the authors demonstrate that their method achieves zero scale-equivariance error.

**Strengths:**

- The paper is well-written, precise, and easy to follow. In particular, their motivation is stated clearly, they introduce the preliminaries succinctly, and the approach stated is coherent and structured.

- To the best of my knowledge the proposed method is novel and the claims that the module modifications achieve scale-equivariance is correct.

- I am unaware of any method but this that solves the chosen problem completely, i.e achieves zero scale-equivariance error when downsampling is done via anti-aliasing and subsampling ('ideal' subsampling). The results are also competent when they do 'non-ideal' subsampling when compared to standard baselines.

- The motivation that images of higher-scale should achieve better classification accuracy than the ones of lower scales is reasonable and the corresponding proposed constraint is sound.

Overall I think the ideas presented in this paper maybe of interest to the broad ML community.

**Weaknesses:**

- Due to the complexity involved in the several DFT/Inverse-DFT related operations, the baselines they compare to which involve kernel resizing are certainly not as costly as the proposed method. I suggest the authors to discuss how their method compares to the baselines in terms of computational/time complexity.

- For the same aforementioned reasons, I suspect that the method might not scale to high dimensional datasets like ImageNet or standard architectures like ResNet50. This I think would be a fundamental limitation, therefore I encourage the authors to discuss the challenges in scaling their method to the mentioned settings. I would be interested to know which modules will be particularly harder to scale.

- Some minor text changes:

     - Line 119, Claim 1: 'even lower in frequencies' needs to be 'even in lower frequencies.'

     - Line 138, Claim 2: 'equivaraint' needs to be 'equivariant.'

- Figure 1, Figtitle: “On the other hand, downsampling the high-res feature is guaranteed to achieve the same low-resolution feature.” -> Unclear formulation. It is not clear how the downsampled version of the features was achieved. The text suggests that it has merely been downsampled from the high-resolution version, but then it is not clear which role the new proposed architecture plays. Please rephrase for clarity.


- Line 32: “Specifically, these works are derived using a continuous domain down-scaling operation, i.e., there is no need to consider anti-aliasing. However, when performing a down-scaling, the Nyquist theorem [21, 27] tells us that an anti-alias filter is necessary to avoid high-frequency content to alias into lower frequencies. (...) To address this gap, from prior work, we consider the down-scaling operation directly in the discrete domain taking into account of the anti-aliasing.” Several grammar errors and the paragraph is unclear to me. In the first sentence, the authors write that previous works “do not need to consider anti-aliasing”. But then, they aim to bridge the gap and take anti-aliasing into account. Please fix the formulation as it is currently very unclear whether previous works had an issue with anti-aliasing or not.


- Line 103: The notation for the definition of g is unclear. It is unclear what is meant by that x is \in {R^1, R^2 … R^N}, which makes it hard to understand the following definitions. The notation of {..} suggests that x can be from R^1, but then how is the Fourier transform of one number even defined? The Fourier transform is needed for D_R(x). I am confused by this definition, please rewrite for clarity.


- It would be helpful if the authors would state that small x refers to the spatial domain and big X always refers to the Fourier domain. While this is the common practice, it would still be nice to state it explicitly.


- I don’t understand Fig. 2b. Are the yellow regions ones and the gray ones zero?


- Line 142: “Element-wise non-linearities, e.g., ReLU, in the spatial domain are generally not scale-equivariant under the ideal downsampling operation DR.” The reasoning of this claim is not obvious to me, please explain / give some intuition.




**Questions:**

I encourage the authors to respond to the points I raised on weaknesses and fix minor text errors. I am willing to raise my score if the authors adequately respond to my concerns.

**Limitations:**

The authors have not included a Broader Impact section and did not discuss the potential negative societal impact of their work. They also do not have a discussion section where some limitations of their work could have been discussed.

---

> ### Author Rebuttal · Authors · 2023-08-09
>
> > Q1. Computational/time complexity.
>
> Please see the general response for a detailed discussion.
>
> > Q2. Discussion on limitation
>
> Please see the general response.
>
> > Q3. Typos
>
> Thanks for the careful review. We will address the typos.
>
> > Q4. Caption clarification for Fig. 1
>
> Fig. 1 illustrates the exact scale equivariance property of the neural network $g$, i.e., $D_R(g(x)) = g(D_R(x))$. Where $D_R$ is the ideal downsampling, representing scaling operation, $g$ is the neural network, and  $g(x)$ is the features of the image $x$. In the caption, the 'high-res features' refers to the extracted feature maps from our approach using a high-resolution image, while the 'low-res features' refers to those from a downscaled, low-resolution image. As our model achieves exact equivariance, the 'low-res' feature precisely matches the downscaled version of the 'high-res' features. We will revise this caption to enhance clarity.
>
> > Q5. Clarification Line 32
>
> The down-scaling operation of a signal $f$ can be written as $D_s\[f\](x) = f(sx)$ with $s>1$. When $f$ is in its continuous form, such an operation does not produce any artifact. On the other hand, when $f$ is sampled and represented as a finite-sized vector, such a down-scaling operation refers to the discarding of values. For example, down-scaling by $s=2$ can be performed by discarding every other sampled point. This naive discarding of sampled values might reduce the sampling rate below the Nyquist Frequencies and introduces the aliasing effect.
>
> In short, aliasing happens only when the signal is sampled, not in its continuous form.  Previous work considers signals in their continuous form thus avoiding the issue of aliasing, which is not practical. As for most practical purposes, we always work with sampled signals. Our proposed method considers the aliasing effect and works directly on sampled signals. We will rewrite it for clarity.
>
>
> > Q5. Clarification Line 103
>
> In Line 103, we define a neural network, $g$,  acting directly on the discretized/sampled signal. When we sample any signal, depending on the sampling rate we get vectors of different sizes. In other words, the domain of the neural network is vectors of different sizes which is defined as the set $\\{ R^1, R^2, … R^N \\}$. Each $R^n$ represents all possible real vectors of size $N$.
>
>
> The output of the neural network will be at the same sampling rate as the input. So, the range of the neural network $g$ is also $\\{ R^1, R^2, … R^N \\}$, i.e., it is composed of spaces of vectors of different sizes. It also can be equivalently described as $\cup_{j=1}^n R^j$.
>
> Side note, the DFT of a single number is just defined to be itself. We will also rewrite the section for clarity.
>
> > Q7. The notation for small $x$ and big $X$.
>
> Yes, we followed the common practice of using small $x$ for the spatial domain and big $X$ for the Fourier domain. As suggested, we will explicitly write this out.
>
> > Q9. The explanation for Fig. 2b. Are the yellow regions ones and the gray ones zero?
>
> The gray regions correspond to zeros (See Fig. 2b’s caption) and the yellow regions correspond to non-zero values. The figure illustrates the intuition that an output's higher frequencies can only depend on even lower frequencies in the input.
>
> > Q10. Intuition for Element-wise non-linearities, e.g., ReLU, being not scale-equivariant under the ideal downsampling operation.
>
> In short, the low-pass filter (smoothing) of the value of the signal at a point during the ideal-downsampling operation depends on the neighboring values. Specifically, smoothing is not commutative with ReLU.
>
> To see this, consider a signal that alternates between +2 and -4, and we use a box filter (a simple average) of size 2 with circular padding for smoothing.
>
> [+2,-4,+2,-4,+2,-4] → Smooth → [-1,-1,-1,-1,-1,-1] → ReLU → [0,0,0,0,0,0]
>
> [+2,-4,+2,-4,+2,-4] → ReLU → [2,0,2,0,2,0] → Smooth → [1,1,1,1,1,1]
>
> Observe that the two outputs are different, which proves that ReLU and smoothing are not commutative.

---

> > ### Comment · Reviewer_JmYY · 2023-08-12
> > **Response to the authors' rebuttal**
> >
> > Dear authors,
> >
> > thank you for responding to my concerns. Most of my questions have been answered. Do you think this approach could be scaled to larger DNNs on larger datasets, such as e.g. a ResNet50 on ImageNet? I appreciate the complexity analysis and the statement in the general response: "From our understanding, modern GPUs are heavily optimized for regular convolution operation which can be done in place. On the other hand, the FFT algorithm does not get the full advantage of GPUs due to reasons like unique memory access patterns, moderate arithmetic intensities, etc." Given we had a narrow-purpose accelerator, such as a parallel processing unit which would be specialized to FFT operations instead of regular convolutions, could this method be straight-forwardly applied to larger DNNs or do you see any other fundamental issues in scaling it?
> >
> > Best,
> > Reviewer JmYY

---

> > > ### Author Response · Authors · 2023-08-13
> > > **Further Discussion with Reviewer JmYY**
> > >
> > > Dear Reviewer JmYY,
> > >
> > > We thank the reviewer for the prompt response.
> > >
> > > Beyond the discussion in the general rebuttal, we do not foresee fundamental issues in scaling our approach, any more than group convolution,  if we are given specialized hardware for FFT. With that said, further study is required to confirm this. For example, it is unclear whether hyperparameters or network architecture can be directly transferred to bigger models and different datasets. However, these are aspects that are not unique to the proposed method.
> > >
> > > Best,
> > >
> > > Authors

---

> > > > ### Comment · Reviewer_JmYY · 2023-08-13
> > > > **Further discussion**
> > > >
> > > > Dear authors,
> > > >
> > > > thank you for your fast response. As indicated in the initial review, I am raising my score to 7 and will advocate for accepting this paper.
> > > >
> > > > Best,
> > > > Reviewer JmYY

---

### Official Review · Reviewer_5utz · 2023-07-05

**Soundness:** 3 good
**Presentation:** 3 good
**Contribution:** 3 good
**Rating:** 6
**Confidence:** 3

**Summary:**

This paper introduces Truly Scale-Equivariant Deep Nets with Fourier Layers, which directly formulate down-scaling in the discrete domain to address the anti-aliasing problem. The proposed methods are validated on the MNIST-scale and STL-10 datasets, demonstrating good classification performance while preserving zero equivariance error.

**Strengths:**

1. This paper addresses how to learn scale-equivariant representations in deep neural networks, overcoming the limitation of previous methods that did not account for anti-aliasing.
2. The paper proposes a method that can consider the down-scaling operation directly in the discrete domain with anti-aliasing and achieves good performance.


**Weaknesses:**

1. This paper does not provide sufficient experiments to demonstrate the generality of the proposed method. It only tests on two small datasets and a single high-level vision task (image classification), which limits the impact of the proposed method. This paper should also evaluate the method on more larger-scale datasets, more high-level vision tasks (e.g., image segmentation), and more CNN architectures.
2. Why does the baseline use zero-padding to the original resolution as the input, instead of resizing to the original resolution? Does this affect baseline's representation learning and performance?
3. The paper lacks an analysis of the computational cost of the proposed method and other compared baseline during training and inference.


**Questions:**

Please refer to the weakness part.

**Limitations:**

Not applicable.

---

> ### Author Rebuttal · Authors · 2023-08-09
>
> > Q1. Evaluation of large-scale datasets and dense prediction tasks.
>
> Thanks for the suggestion. For a consistent evaluation, we followed prior works on scale-equivariance to evaluate using the MNIST and STL10 datasets, e.g., [36].
>
> We agree that large-scale datasets and dense prediction tasks would be interesting. However, computation and memory bottlenecks remain challenging for our method **and** the compared scale-equivariant methods. We will add a limitation section to discuss this open challenge.
>
> Next, extension to semantic segmentation requires finding a suitable architecture.  The existing spatial domain segmentation architectures are not directly applicable to perfect scale equivariant operations and Fourier layers.
>
> For example, the popular semantic segmentation architecture U-Net [i] or its neural operator variant [ii] contains downsampling, skip-connections, and upsampling, which first reduce the size of the input image and then expand the output to the original resolution. In other words, a scale-equivariant, with consideration of anti-aliasing, upsampling layer is needed to undo the effect of downsampling layers. However, as the scaling operation on discrete vectors forms a semi-group, it does not necessarily contain inverse operations. It remains an open challenge to design truly scale equivariant Fourier layers that support upsampling operations. We plan to investigate this problem in the future.
>
> Finally, our experiments aim to validate the effectiveness of our model on the benchmark datasets of other scale-equivariant techniques. We also conducted ablations to justify each of the proposed components. We believe the empirical evaluation is sufficient to validate our contribution. In future work, we aim to extend the proposed technique to larger datasets and dense prediction tasks.
>
> - [i] Ronneberger, Olaf, Philipp Fischer, and Thomas Brox. "U-net: Convolutional networks for biomedical image segmentation." In Proc. MICCAI, 2015
> - [ii] Rahman, Md Ashiqur, Zachary E. Ross, and Kamyar Azizzadenesheli. "U-no: U-shaped neural operators." arXiv preprint arXiv:2204.11127 (2022).
> - [36] Sosnovik, Ivan, Michał Szmaja, and Arnold Smeulders. "Scale-equivariant steerable networks." In Proc. ICLR, 2022
>
> > Q2. Why does the baseline use zero-padding to the original resolution as the input, instead of resizing to the original resolution? Does this affect the baseline's representation of learning and performance?
>
> The use of zero-padding follows strictly from prior works [8, 12, 36, 37].
> In the supplemental, we include results on ablating zero-padding vs. resizing the image (See Tab. A2 and Tab. A4 in the appendix). Baselines show little improvement in their classification accuracy, whereas the equivariance error does not show any improvement.
>
> - [8] R. Ghosh and A. K. Gupta. Scale steerable filters for locally scale-invariant convolutional neural networks. arXiv preprint arXiv:1906.03861, 2019.
> - [12] A. Kanazawa, A. Sharma, and D. Jacobs. Locally scale-invariant convolutional neural networks. arXiv preprint arXiv:1412.5104, 2014.
> - [36] I. Sosnovik, M. Szmaja, and A. Smeulders. Scale-equivariant steerable networks. In Proc. ICLR, 2020
> - [37] I. Sosnovik, A. Moskalev, and A. Smeulders. DISCO: accurate discrete scale convolutions. In Proc. BMVC, 2021
>
> > Q3. Comparison of computation or memory usage
>
> Please see the general response for a detailed discussion.

---

> > ### Comment · Reviewer_5utz · 2023-08-13
> > **Response to the authors' rebuttal**
> >
> > Dear authors,
> >
> > Thank you for responding to my questions. My concerns have been well addressed. I would like to increase my rating to Weak Accept. The reason why I don't give a higher score is that currently, this method still has computation and memory bottlenecks.
> >
> > Best regards,
> >
> > Reviewer 5utz

---

> > > ### Author Response · Authors · 2023-08-13
> > > **Further Discussion with Reviewer 5utz**
> > >
> > > Dear Reviewer 5utz,
> > >
> > > We are glad that the concerns have been addressed. As mentioned in the general response, we will include a section discussing the computation and practical implications. Please let us know if further questions arise. We would be happy to respond.
> > >
> > > Best,
> > >
> > > Authors

---

### Official Review · Reviewer_JFCH · 2023-07-05

**Soundness:** 3 good
**Presentation:** 3 good
**Contribution:** 3 good
**Rating:** 6
**Confidence:** 3

**Summary:**

The submitted paper explores the problem of scale-equivariant neural networks.
To get the desired property by design, the authors propose new building blocks: a spatially local Fourier layer, a novel scale-equivariant pooling layer, and scale-equivariant way of applying activation functions.
All the introduced modifications rely heavily on the Fourier transform of the input feature map.
Additionally, a new loss function is proposed which aims to promote the natural consistency of predictions made at different scales.

The effect of the new approach is demonstrated on the problem of image classification on two datasets: MNIST-scale and STL-10-scale.

**Strengths:**

I find the proposed modifications well explained and motivated.
To the best of my knowledge, the presented approach is original enough, and it may be interesting for the broad community.
Quality of the results seems very satisfactory and I hope they can serve as a reliable baseline for further research on this topic.


**Weaknesses:**

1. While the proposed results seem strong enough for the task of object recognition, the authors have not reported the computational or memory overhead introduced by their approach. Without this knowledge, it is hard to understand if their method can be directly scaled to more high-resolution data without any significant changes.

1. I admit that most prior papers also mostly report their performance for image classification, it is quite clear that the property of scale equivariance may be even more desirable for the dense prediction tasks, e.g. semantic segmentation. Are there any barriers to applying the same method to the common segmentation architectures?

1. Regarding the low performance of baseline models on STL-10-scale data (line 274), have the authors tried other padding schemes than zero-padding, e.g. so-called reflect/replicate/circular strategies (see the [link](https://pytorch.org/docs/stable/generated/torch.nn.functional.pad.html))? I find it suspicious that, e.g. the reported DISCO performance of 91.93% accuracy on STL-10 [1] drops to 47.68% on STL-10-scale (Tab. 4) in the provided evaluation results. I think, more discussion is needed on that matter.

[1] Sosnovik et al. DISCO: accurate Discrete Scale Convolutions. In BMVC, 2021.

**Questions:**

Please, address the aforementioned weaknesses.

**Limitations:**

No additional inquiries.

---

> ### Author Rebuttal · Authors · 2023-08-09
>
> > Q1. Computational or memory overhead of the approach
>
> Please see the general response for a detailed discussion.
>
> > Q2. Evaluation of dense prediction tasks, e.g., semantic segmentation?
>
> For a consistent evaluation, we followed prior works on scale-equivariance to evaluate using the MNIST and STL10 datasets, e.g., [36]. We note that extension to semantic segmentation requires finding a suitable architecture. Existing spatial domain segmentation architectures are not directly applicable to scale equivariant operations with Fourier layers.
>
> For example, the popular semantic segmentation architecture U-Net [i] or its neural operator variant [ii] contains downsampling, skip-connections, and upsampling, which first reduce the size of the input image and then expand the output to the original resolution. In other words, a scale-equivariant, with consideration of anti-aliasing, upsampling layer is needed to undo the effect of the downsampling layers. However, as the scaling operation on discrete vectors forms a semi-group, it does not necessarily contain inverse operations. It remains an open challenge to design truly scale equivariant Fourier layers that support upsampling. We plan to investigate this problem in the future.
>
>
> - [i] Ronneberger, Olaf, Philipp Fischer, and Thomas Brox. "U-net: Convolutional networks for biomedical image segmentation." In Proc. MICCAI, 2015
> - [ii] Rahman, Md Ashiqur, Zachary E. Ross, and Kamyar Azizzadenesheli. "U-no: U-shaped neural operators." arXiv preprint arXiv:2204.11127 (2022).
> - [36] Sosnovik, Ivan, Michał Szmaja, and Arnold Smeulders. "Scale-equivariant steerable networks." In Proc. ICLR, 2022
>
>
> > Q3. Low performance of baseline models on STL-10-scale data (line 274). Have the authors tried other padding schemes than zero-padding, e.g. so-called reflect/replicate/circular strategies?
>
> We experimented with STL10-Scale (following the same setup as MNIST-Scale) which contains training images of resolutions 48x48-97x97. We note that the baselines, such as DISCO, do not consider anti-aliasing in their formulation. At large downsampling rate anti-aliasing matters.
>
> Next, for the zero-padding processing step, we followed the prior work’s choice from MNIST-Scale. As suggested by the reviewer, we conduct additional experiments (see table below) using different padding techniques (reflect, replicate, and circular). We observed that the padding schemes do not significantly change the model performance. There is no specific padding scheme that helps all the methods.
>
> | Method \\ Padding | Replicate | Circular | Reflect |
> |----------------|-----------|----------|---------|
> | DISCO          | 0.60      | 0.52     | 0.44    |
> | SESN           | 0.64      | 0.65     | 0.50    |
> | DSS            | 0.61      | 0.48     | 0.49    |
> | SS-CNN         | 0.47      | 0.50     | 0.50    |
> | SI-ConvNet     | 0.54      | 0.63     | 0.63    |

---

> > ### Comment · Reviewer_JFCH · 2023-08-18
> >
> > I would like to thank the authors for their feedback. I keep the original score.

---

### Official Review · Reviewer_qJeg · 2023-07-14

**Soundness:** 4 excellent
**Presentation:** 4 excellent
**Contribution:** 3 good
**Rating:** 7
**Confidence:** 4

**Summary:**

This paper proposes a formulation of scale equivariant networks, which takes anti-aliasing into account. This is in contrast to previous works, in which the issue of aliasing is often disregarded. The proposed method achieves competitive classification accuracy on MNIST-scale and STL-10, while maintaining zero equivariance error.

**Strengths:**

This work addresses a problem that has been thus far disregarded when talking about scale equivariance: the aliasing issue. The proposed methods and solutions are sound and are empirically evaluated with very good results.

**Weaknesses:**

* My main concern is that, to the best of my understanding, the proposed framework does not consider inter-scale relationships. Instead, it considers multiple scales in parallel which are processed independently of one another. This brings into question whether this work should be defined as a work in group equivariance or rather a work on models aiming to be resolution-agnostic, i.e., able to work regardless of the resolution of the input, e.g., FlexConv [1], S4ND[2], \inf-Diffusion [3].

* This method relies heavily on using the Fourier and inverse Fourier transform for each of its layers. In addition, it is well-known that performing convolutions on the spatial domain is faster and requires less memory than performing them in the Fourier domain when convolutional kernels are small O(Nk) vs O(NlogN). However, the proposed method actually relies on small convolutional kernels. These two factors call into question the computational and memory complexity of the proposed method. I believe that this is the main practical limitation of the method, and I would like the authors to discuss this in the paper –in case it is indeed an issue–.

* Connecting to the previous point, I believe that having a section in which the limitations of the work are clearly stated and discussed is vital.

**Questions:**

* I find the definition of the convolutional kernels (Eq. 14) too convoluted. I would simply say that you define a kernel of size k in the spatial domain and send it to the Fourier domain next.

* There are several works that tackle the issue of downsampling breaking equivariance (even in a general group setting) as well as tackling aliasing in many other settings, e.g.,[1,2,3,4,5,6,7] among many more. I believe that discussing these works would be beneficial to scope the paper better.

* In line 213, you mention that you do the analysis of Sosnovik et al on trained models. Although I understand your reasoning, I actually believe that treating non-trained networks directly, i.e., random nets, makes more sense in order to evaluate the real capability of the networks to remove aliasing and preserve equivariance. Have you done this analysis as well? What were the results? Did you still get zero equivariance error?

* I believe line 165 has an error.

* The paper has some small typos. Please correct them.


**Limitations:**

As mentioned previously, the authors have not listed any limitations of their work in the paper.

### Conclusion

In conclusion, I think this is an interesting paper with important contributions to the field. However, I am not sure if this paper should be framed as a scale-equivariant paper or rather as a resolution-agnostic neural architecture. I believe –please do tell me if you don't agree– that the latter option is more accurate. Although it requires some rewriting, I believe it would be possible to still do this during the rebuttal period.

#### References

[1] Romero, D. W., Bruintjes, R. J., Tomczak, J. M., Bekkers, E. J., Hoogendoorn, M., & van Gemert, J. C. (2021). Flexconv: Continuous kernel convolutions with differentiable kernel sizes. arXiv preprint arXiv:2110.08059.

[2]  Nguyen, Eric, Karan Goel, Albert Gu, Gordon Downs, Preey Shah, Tri Dao, Stephen Baccus, and Christopher Ré. "S4nd: Modeling images and videos as multidimensional signals with state spaces." Advances in neural information processing systems 35 (2022): 2846-2861.

[3] Bond-Taylor, S., & Willcocks, C. G. (2023). $\infty $-Diff: Infinite Resolution Diffusion with Subsampled Mollified States. arXiv preprint arXiv:2303.18242.

[4] Karras, Tero, Miika Aittala, Samuli Laine, Erik Härkönen, Janne Hellsten, Jaakko Lehtinen, and Timo Aila. "Alias-free generative adversarial networks." Advances in Neural Information Processing Systems 34 (2021): 852-863.

[5] Zhang, R., 2019, May. Making convolutional networks shift-invariant again. In International conference on machine learning (pp. 7324-7334). PMLR.

[6] Xu, J., Kim, H., Rainforth, T. and Teh, Y., 2021. Group equivariant subsampling. Advances in Neural Information Processing Systems, 34, pp.5934-5946.

[7] Rojas-Gomez, Renan A., Teck-Yian Lim, Alex Schwing, Minh Do, and Raymond A. Yeh. "Learnable Polyphase Sampling for Shift Invariant and Equivariant Convolutional Networks." Advances in Neural Information Processing Systems 35 (2022): 35755-35768.

---

> ### Author Rebuttal · Authors · 2023-08-09
>
> > Q1. Group equivariance vs. resolution agnostic.
>
> We believe our work covers both aspects, where we construct equivariant features to build a classifier that is resolution agnostic.
>
> First, we view our proposed method (Sec. 4.1) to be a method for learning equivariant features. Following the definition of equivariance (Line 103 ), a deep-net $g$ is equivariant with the group action $D_r$ if it commutes with the group action $D_r$, i.e., $D_r(g(x)) = g(D_r(x))$ is satisfied. We proved in the paper that our proposed model is equivariant when $D_r$ is an ideal down_scaling operation (Eq. 6).
>
> Next, in Sec. 4.2, we propose a classifier for using these proposed equivariant features with the goal that the model works well across different input resolutions. Hence, this work also covers the resolution agnostic aspect.  We note that the agnostic approaches like S4ND, and \inf-Diff use a continuous kernel for convolution which we showed that such methods are not truly scaled equivariant when considering anti-aliasing and non-linearity (See “Fourier CNN” in Tab. 1 and Tab. 2).
>
> Finally, we want to clarify that while our approach models the scales in parallel they are not independent of one another. The interactions between the scales happen in two ways:  (a) as stated in Claim 1, the “output’s frequency terms can only have dependencies on the terms in $X$ that are even lower in frequencies.” The lower frequencies can be interpreted as a smaller scaler; (b) the model parameters are shared between images across all scales.
>
> Thanks for this clarifying question, we incorporate our responses here into the paper.
>
> > Q2. Computational and memory complexity of the method
>
> We agree with the reviewer that, when compared with regular CNN, convolution using FFT is not as efficient. However, we think that our work should be compared to group convolution instead of regular convolutions. Please see the general response for a detailed discussion.
>
> > Q3. A limitation section?
>
> Please see the general response.
>
> > Q4. Definition of the convolutional kernels in Eq. 14
>
> Thanks for the suggestion. We derived the kernel analytically to have a better understanding of its form. In hindsight, it indeed seems convoluted. We will address this.
>
> > Q5. Related works on downsampling breaking equivariance
>
> Thanks for the list of related works regarding anti-aliasing breaking other equivariance. In the paper, we focused on reviewing related works on scale-equivariance. We will include and discuss all the mentioned papers.
>
> > Q6. Evaluation of equivariance with random initialization
>
> We also understand the perspective of evaluating with random initialization. Yes, our model achieves zero equivariance error even when tested with randomly initialized parameters. We will add this result to the paper.
>
> > Q7. Small typos.
>
> Thanks for the detailed review. We will address this.
>
> > Q8. Rewriting during the rebuttal period.
>
> It seems like NeurIPS does not allow updates to the paper during the rebuttal period. We find the suggestions to be very helpful and will incorporate all the recommended changes.

---

> > ### Comment · Reviewer_qJeg · 2023-08-13
> > **Response to Rebuttal**
> >
> > Dear authors,
> >
> > Thank you for responding to my questions. Your answers have addressed my concerns. As far as these answers are reflected in the camera-ready version of the paper, I am happy to support acceptance.
> >
> > Just as a final observation, regarding this response:
> > > The interactions between the scales happen in two ways: (a) as stated in Claim 1, the “output’s frequency terms can only have dependencies on the terms in that are even lower in frequencies...”
> >
> > I indeed agree with the authors. However, I would like to note that in papers such as SESN, B-Spline G-CNNs, Wavelet Networks [2], Deep Scale Spaces (just to name a few), scale group convolutions can depend on all scales considered at every layer, i.e., not only on lower frequencies. Perhaps your method is actually better than making the group conv depend on all scales, as it avoids padding along the scale axis (which is problematic as mentioned by these methods), and simultaneously addresses aliasing. Perhaps a small analysis on this could also add to the paper.
> >
> > Best regards,
> >
> > Reviewer qJeg
> >
> > [1] Romero, D.W., Bekkers, E.J., Tomczak, J.M. and Hoogendoorn, M., 2020. Wavelet networks: Scale equivariant learning from raw waveforms. arXiv preprint arXiv:2006.05259.

---

> > > ### Author Response · Authors · 2023-08-13
> > > **Further Discussion with qJeg**
> > >
> > > Dear Reviewer qJeg,
> > >
> > > We will include all the discussed materials and suggestions in the final version.
> > >
> > > We thank you for the thought-provoking question regarding the scale dependencies and padding. We will look into these aspects, e.g., analyzing (empirically and analytically) the baselines' frequency response.
> > >
> > >
> > > Best,
> > >
> > > Authors

---

### Author Rebuttal · Authors · 2023-08-09

We thank the reviewers for their constructive feedback and appreciation of our contribution. We first address the common concerns. Individual questions are answered separately in each reviewer's thread.

> Q1. Computational and memory complexity of the method

We now provide further analysis and empirical results regarding memory and computational complexity.

Consider a 1D signal of length $N$ and a kernel size of $K$. Our proposed model involves: (a) a transformation from local filter to global filter with a time complexity of $O(KN)$, (b) a convolution using Fourier transform with a time complexity of $O(N \log N)$, and (c) our scale equivariant non-linearity where the complexity depends on the group size. Let $A$ be the set of group actions, then the time complexity of the proposed scale-equivariant non-linearity is $O(|A| N \log N)$, where $|A|$ denotes the cardinality of the set $A$. The time complexity for each layer is then $O(|A| N \log N + KN)$.

As a comparison, the time complexity of regular a group convolution layer [i] is $O(KN|A|)$ in the first layer and $O(KN|A|^2)$ for all intermediate layers [ii], assuming the cost of group action is a negligible constant.

Considering the time complexity of the intermediate layers of group convolutions, our proposed method is more efficient when
$|A| N \log N + KN < |A|^2KN$
$\implies \log N + K/|A| < |A|K$.

In other words, if $K << |A|$ and $\log N < |A|K$, i.e., assuming the set of group actions of moderate size, then our method is faster than group convolutions.

- [i]. Cohen, Taco, and Max Welling. "Group equivariant convolutional networks." Proc. ICML, 2016.
- [ii]. He, Lingshen, et al. "Efficient equivariant network." Proc. NeurIPS, 2021


Next, we present the measured run time and memory usage of a single-scale equivariant convolution layer for different methods (See the table below). These results are measured using a batch of 200 images of resolution $28 \times 28$. Please note that runtime and memory usage for our approach are measured including the scale-equivariant non-linearity. Other approaches use regular pointwise non-linearities that are done in place with no memory overhead.

**Number of Parameters, run time, and memory requirements:**
| model      | #parameter | Exe. Time on CPU (sec) | Memory (MB) | Exe. Time on GPU (sec) |
|------------|------------|------------------------|-------------|-----------------------|
| Disco      | 50272      | 0.857                  | 0.77        | 0.0008                |
| SS-CNN     | 56384      | 0.220                  | 0.05        | 0.0070                |
| SI-ConvNet | 50272      | 0.200                  | 0.24        | 0.0010                |
| SESN       | 50272      | 0.304                  | 0.96        | 0.0004                |
| DSS        | 50240      | 0.162                  | 0.96        | 0.0012                |
| Ours       | 50496      | 0.071                  | 0.59        | 0.0360                |

We observe that our approach requires less memory than all other scale-equivariant approaches except for SS-CNN. Our approach’s runtime on the CPU is lower than other approaches, however, this reduction does not transfer to GPU. From our understanding, modern GPUs are heavily optimized for regular convolution operation which can be done in place. On the other hand, the FFT algorithm does not get the full advantage of GPUs due to reasons like unique memory access patterns, moderate arithmetic intensities, etc. As there are many efforts to accelerate FFT for GPUS [iii, iv], we believe this performance gap will be reduced in the future. We will include all these results in the paper and add a limitation section.

- [iii]. Durrani, Sultan, et al. "FFT blitz: the tensor cores strike back." In Proc. ACM SIGPLAN, 2021.
- [iv]. Li, Binrui, Shenggan Cheng, and James Lin. "tcfft: Accelerating half-precision FFT through tensor cores." arXiv, 2021.

>Q2. Limitation & Broader Impact Section

Thanks for the suggestion. Yes, we agree that a limitation section will strengthen the paper. We will include a discussion on the limitation of the computation and the practical implications. Next, we do not foresee any negative societal impact caused by our work. We conducted experiments on image classification using standard benchmarks. These benchmarks are widely studied by the machine learning and computer vision community.

---

### Decision · Program_Chairs · 2023-09-21

**Decision:**

Accept (poster)

**Comment:**

This work closely studies a particular aspect of signal processing for achieving scale equivariance in convolutional nets and demonstrates improvements in task accuracy, consistency across scales, and the equivariance of feature maps across scales. These results rely on the the proposed Fourier filtering layer and accompanying scale-equivariant nonlinearity and pooling. Four expert reviewers with backgrounds in filtering, equivariance, and generalization agree on acceptance (qJeg: 7, JmYY : 7, JFCH: 6, 5utz: 6). While the experiments are limited to toy datasets (MNIST, STL) for the analysis of spatial scale, these datasets are inherited from prior work, and the comparison with existing works is comprehensive and fair. The experiments in ideal and non-ideal filtering conditions, reflecting realistic computation, demonstrate the reduction of scale equivariance error to zero and least w.r.t. baselines respectively. Therefore, while this work could be more broadly impactful if larger models and data were considered, it is still an improvement within its scope. Its signal processing contribution to deep convolutional nets can inform the community for the pursuit of spatial scale equivariance on more tasks and larger scales of models and data.

For clarity, the AC advises the authors to 1. mention the limitation—shared by this work and others—that equivariance is restricted to a chosen set of discrete and bounded resolutions and 2. reference the supplement from the text for the results on resizing vs. padding (see comment by 5utz) to verify the fairness of these comparisons.